# Contribution of Trunk Muscles to Upright Sitting with Segmental Support in Children with Spinal Cord Injury

**DOI:** 10.3390/children7120278

**Published:** 2020-12-08

**Authors:** Goutam Singh, Sevda Aslan, Beatrice Ugiliweneza, Andrea Behrman

**Affiliations:** Department of Neurological Surgery, University of Louisville, Louisville, KY 40202, USA; goutam.singh@louisville.edu (G.S.); sevda.aslan@louisville.edu (S.A.); beatrice.ugiliweneza@louisville.edu (B.U.)

**Keywords:** trunk muscle activation, spinal cord injury, children, surface electromyography

## Abstract

To investigate and compare trunk control and muscle activation during uncompensated sitting in children with and without spinal cord injury (SCI). Static sitting trunk control in ten typically developing (TD) children (5 females, 5 males, mean (SD) age of 6 (2)y) and 26 children with SCI (9 females, 17 males, 5(2)y) was assessed and compared using the Segmental Assessment of Trunk Control (SATCo) test while recording surface electromyography (EMG) from trunk muscles. The SCI group scored significantly lower on the SATCo compared to the TD group. The SCI group produced significantly higher thoracic-paraspinal activation at the lower-ribs, and, below-ribs support levels, and rectus-abdominus activation at below-ribs, pelvis, and no-support levels than the TD group. The SCI group produced significantly higher lumbar-paraspinal activation at inferior-scapula and no-support levels. Children with SCI demonstrated impaired trunk control with the ability to activate trunk muscles above and below the injury level.

## 1. Introduction

Trunk muscles innervated by the thoracic and lumbar spinal nerves are critical for stability and mobility as they are involved in virtually all movements requiring an upright posture whether seated, standing, or walking [1,2,3]. The development of independent sitting is accomplished during the first year of life, as infants gradually learn to maintain their center of mass within the base of support. This is achieved by activating trunk muscles to hold the trunk upright during independent sitting and reaching tasks [4,5]. As children continue to develop, the activation and coordination of trunk muscles are required for more dynamic sitting and the transition to standing and walking [6,7]. Therefore, adequate trunk control in children is critical to participate in their environment. 

Spinal cord injury (SCI) in children at cervical and upper thoracic cord levels results in paralysis or paresis of trunk muscles, as well as muscles of upper and lower extremities. As a consequence, children are unable to maintain an upright posture during sitting [1,8,9]. Compared to adults, paralysis of trunk muscles in children has an added complexity of on-going musculoskeletal growth through childhood and adolescence, placing children at greater risk for developing scoliosis [10,11]. The International Standards for Neurological Classification of Spinal Cord Injury (ISNCSCI) is used by therapists to assess the spinal cord motor and sensory function and then classify the severity of SCI [12] via the American Spinal Injury Association Impairment Scale (AIS). The scale, however, uses the outcomes of sensory perception with the patient tested in the supine position to indirectly assess the motor function of the trunk muscles [13,14]. The AIS scale is unreliable for assessing children under six years of age due to the inability to fully comprehend and follow instructions during the test [15]. Lack of appropriate tools to evaluate sitting trunk control has been a challenge for its assessment, particularly in children following SCI. Without such instruments, developing effective treatments and strategies also is difficult. Using surface electromyography (EMG) in a seated position, may be an effective alternative to the AIS to examine the activation of trunk muscles in adults, [16] adolescents, and children with SCI. 

Tests such as the Trunk Impairment Scale and the Gross Motor Function Classification System have been used to measure trunk control in children and adults with neurological based impairments [17,18]. However, independent sitting and standing by participants is a prerequisite for these tests. Therefore, testing trunk control in children who have not achieved independent sitting and in children with a low functional level is limited [19]. In addition, these tests measure trunk performance as a single unit with trunk often collapsing in a “c” posture, i.e., compensation. A new pediatric, measurement instrument, the Segmental Assessment of Trunk Control (SATCo), was recently introduced and validated to assess and track improvements in trunk control in children with SCI lacking independent sitting or impaired sitting control [20]. The SATCo is based on observation of the natural acquisition of trunk control in typically developing (TD) children which occurs segmentally in a cephalocaudal direction and has been used in children with neuromuscular disorders [21]. The test uniquely assesses posture control in the sitting position with a stable, neutral base of pelvis controlled via straps or manually with support. External support is provided in a segmental manner at each biomechanical level: shoulders, axillae, inferior-scapulae, ribs, below-ribs, pelvis, and finally no-support while trunk control, above the external support, is evaluated for static, active, or reactive control. For evaluation of static control, a neutral vertical trunk position in the sagittal and frontal plane is held for five seconds as the individual maintain the head in midline [22]. 

We undertook this study as a critical step to understand the contribution of trunk muscles during upright sitting in children with SCI. The aim, thus, of this study was to examine and compare levels of trunk control per the SATCo and trunk muscle activation during uncompensated sitting in children with SCI to TD children. We selected to test children during the static control component, i.e., quiet sitting, while simultaneously recording trunk muscle activity. We hypothesized that children with SCI would show (a) significant deficits in trunk control measured by SATCo scores compared to age-matched TD children and (b) significantly lower or absent trunk muscle activation below the injury level at the same segmental support level compared to age-matched TD children. 

## 2. Materials and Methods

### 2.1. Participants and Clinical Characteristics 

The Institutional Review Board (IRB) at the University of Louisville approved this study (IRB protocol #15.0585). Informed consent and assent were signed by legal guardians of participant and participant above 7 years of age, respectively. Ten TD participants (5 females and 5 males, age 6 ± 2 years) (Table 1) were recruited from the community. The Human Locomotion Research Center Database at the University of Louisville, Louisville, KY, USA (IRB approval # 06.0647), was used to recruit twenty-six participants with chronic traumatic or non-traumatic SCI (9 females and 17 males, age 5 ± 2 years). The severity and level of SCI in children of 6 years of age or above were classified using the AIS by ISNCSCI and children below 6 years of age were classified as listed in their medical records.

#### 2.1.1. Trunk Control Assessment

Trunk control was assessed using the SATCo test. Participants were tested in a seated position with arms and back unsupported on a bench with hip and knee both at 90° of flexion with feet supported. Trunk control was examined with a therapist progressively changing the level of support from top at shoulder girdle and axilla to assess cervical (head) control, inferior-scapula (mid-thoracic), lower-ribs (lower thoracic), below-ribs (upper-lumbar), pelvis (lower-lumbar), and no-support, to measure full trunk control [22]. The pelvis was placed in a neutral position via manual support, except for no support level [20]. The testing for static control was successful if participants were able to maintain a stable and upright trunk for at least 5 seconds above the level of support without compensatory strategies such as neck or arm extension, trunk hyperextension (use of reflex activity or spasm) or excessive trunk lordosis [20]. If the participant exhibited compensatory strategies, the test allowed for correction, if compensation persisted, i.e., failed, the test ended, and the prior successful test score was reported as the final score and level. 

#### 2.1.2. Surface Electromyography (EMG)

During the SATCo test, EMG signals from trunk muscles were recorded bilaterally using wireless, pre-amplified bipolar electrodes (Cometa, Italy). Skin area over the muscle belly was cleaned with alcohol swabs and electrodes were placed on the following muscles: pectoralis major (PEC) at midclavicular line; external intercostal (INT) at 6th intercostal space on axillary line; rectus abdominus (RA) at umbilical level; external oblique (OB) at midaxillary level; thoracic paraspinal (PST) at T9–10 vertebral level; and lumbar paraspinal (PSL) at L4–5 vertebral level [23]. The EMG signals were sampled at 2000 Hz with a bandpass filter of 10–500 Hz. Root mean square (RMS) values of each trunk muscle activity acquired during SATCo performance and over a 5 second window were calculated. 

#### 2.1.3. EMG Normalization

Root mean square values of each muscle were calculated using the first SATCo level and used as the baseline level of muscle activation level for both the groups since participant’s trunk and arms were fully supported. EMG signals were normalized by subtracting the baseline EMG from EMG signals recorded during progressive levels of SATCo. Therefore, the two groups were compared for rest of the levels.

### 2.2. Statistical Analyses

The continuous variables (age and BMI) were summarized using median with interquartile range (25th and 75th percentile). Categorical variables (sex and injury-level) were summarized using frequency count with associated percentage (Table 2). Each participant in the TD group scored 20/20 on SATCo test therefore, a one sample median test was used to compare the variable SATCo scores of participants in the SCI group to standard/fixed scores in the TD group. Normalized RMS values for each muscle at each SATCo level were presented as the median with interquartile range (Table 3). Values of the group with SCI and TD were compared with the Wilcoxon Rank Sum test for the central tendency. Statistical analyses were performed in SAS (SAS Institute, NC) and graphical displays were plotted in R.

## 3. Results

### 3.1. SATCo Scores Comparison between TD and Participants with SCI

The SCI group scored significantly lower (*p* < 0.001) on the SATCo compared to the TD group. The SCI group demonstrated impaired trunk control on the SATCo: 10 ± 3 (Mean ± SD) with no child demonstrating the highest score possible, 20. While every participant in the TD group scored, as anticipated, the full score of 20 for trunk control (Figure 1).

### 3.2. EMG Muscle Activation between TD and Participants with SCI during Static SATCo Events

Every participant in the TD group (20/20) completed all levels of SATCo test, whereas many participants in the SCI group (21/26) failed to complete all levels of SATCo testing. All 26 participants in the SCI group successfully completed axilla and scapular levels, 23/26 completed over-lower-ribs, 16/26 completed below-ribs, 12/26 completed pelvis, and only 5/26 completed no-support level. Therefore, the data analyses were presented only for participants in the SCI group who successfully performed static control at each support level compared to the performance by participants in the TD group at the same level (Figure 2).

### 3.3. Paraspinal Muscle Activation

No significant differences in PST muscle activation was observed between the groups at axilla, inferior-scapula, and no-support levels (Figure 2a). Starting with the over-lower-ribs support level to each sequential level, the SCI group had a significantly higher PST muscle activation than the TD group (over-lower-ribs: *p* = 0.01, below-ribs: *p* = 0.001, and pelvis: *p* = 0.03). No significant differences in PSL muscle activation was observed between the two groups at axilla, over-lower-ribs, below-ribs, and pelvis level (Figure 2b). However, the SCI group produced significantly lower EMG magnitude at inferior-scapula (*p* = 0.03) and significantly higher EMG magnitude at no-support (*p* = 0.004) level compared to the TD group (Table 3).

### 3.4. Abdominal Wall Muscles

For the first three levels, i.e., axilla to over-lower-ribs, RA muscle activation was not significantly different between the two groups. From below-ribs to each sequential level, the SCI group had significantly higher RA muscle activation than the TD group (below-ribs: *p* = 0.02, pelvis: *p* = 0.003, and no-support: *p* = 0.007) (Figure 2c). For the first four levels, i.e., axilla to below-ribs, no significant differences in OB muscle activation was found between the two groups (Figure 2d). For the last two levels, pelvis and no-support, the SCI group produced significantly higher OB muscle activation than the TD group (pelvis: *p* = 0.002 and no-support: *p* = 0.009).

### 3.5. Upper Thorax Muscles

A significantly higher PEC muscle activation (*p* < 0.001) was produced in the SCI group at below-ribs compared to the TD group (Figure 2e). The INT muscle activation only at the inferior-scapula level was significantly higher (*p* = 0.04) in the TD group compared to the SCI group (Figure 2f). 

## 4. Discussion

The main findings of this study were, first, that children with SCI, expectedly, demonstrated significant impairment in trunk control assessed by the SATCo. Second, unexpectedly, children with SCI activated trunk muscles not only above the injury level, but also below the injury level. 

Each child in the TD group completed the test with highest possible score (total of 20). While, 1/26 achieved the highest score of 19 and 2/26 achieved highest score of 18. In the SCI group, only 5 out of 26 children completed the no-support static level of SATCo successfully. This significant difference in SATCo scores in children with SCI was expected due to weakness or paralysis of trunk muscles resulting in the inability to sit upright or sit with impaired posture when challenged, yet segmental support was provided [20]. In parallel to the assessment of trunk control, we examined and compared trunk muscle activation using EMG in children with SCI to TD children. EMG assessment in both, TD and SCI groups, allowed objective quantification of trunk muscle activity with each level of support. We found that children in both the groups, produced activation in all six trunk muscles, recorded above and below the level of support (Figure 2). In addition, children in the SCI group, unexpectedly, produced higher activation in trunk muscles below injury level (Figure 2). This quantification of trunk muscle activity using EMG is of significance, as it allows characterization of the extent of sensorimotor function below injury level which may persist following SCI in children (Figure 3). 

### 4.1. Activity Level in Paraspinal Muscles

Erector spinae muscles are primary extensors of the trunk and considered postural muscles, as they are active during sitting, standing upright and during trunk extension tasks [24]. We found that children in both groups, TD and SCI produced activation in both lumbar and thoracic paraspinal muscles at every support level during SATCo. However, the activity level in the two groups was significantly different with children in the SCI group producing higher activation with decreasing level of support from over-lower-ribs to no-support level. Interestingly, all children in the SCI group produced activation in thoracic paraspinal muscles irrespective of injury level (Figure 2a). In addition, several children with SCI also produced activation in lumbar paraspinal muscle below injury level. However, the muscle activity was significantly higher only for those successfully performing the static, no-support level (Figure 2b). Thoracic-paraspinal muscle activity recorded at T10-T11 spinal level and lumbar at L4-L5 vertebral level corresponds to T11-T12 and L4-L5 spinal segment, respectively [25]. In the SCI group, 25/26 children were injured at or above T12 level and 1/26 was injured at L1 level (Table 1). This increased muscle activation in the SCI group indicates the potential to recruit available erector spinae muscle to promote upright sitting. While only 5 children in the SCI group successfully completed the last level of SATCo, each child produced activation in paraspinal muscles (Figure 3). These findings support the presence of residual motor activity below the injury level. Additional factors, such as contraction of antagonistic muscle, spasm or reflexive activity may facilitate/influence activation of trunk muscles below the injury level [8]. However, in the current study, we excluded any postural attempts that may have resulted due to spasms or compensation. Recent work from our laboratory quantified pattern and activity of trunk and lower limb muscles during a sit-up task in children with SCI and reported similar findings with 24 children with SCI produced activation in muscles above and below injury levels. Though the task was performed in supine position [26]. Studies in adults with SCI reported similar results, where individuals produced activation in trunk muscles, specifically in paraspinal muscle at T5 and T12, below injury level [8,16]. In our study, in addition to the presence of activation in paraspinal muscles below injury level, we also observed that the activation was significantly higher than the TD children.

### 4.2. Activity Level in Abdominal Wall Muscles

Numerous studies have shown that abdominal wall muscles, rectus-abdominus and external-oblique muscles, contribute to the stability of the trunk and in preparation for limb movements [27,28]. Interestingly, children with SCI produced significantly higher activation in the RA muscle as the support level was moved from below-ribs to no-support (Figure 2c). Whereas OB muscle activation was significantly higher in the SCI group for last two support levels, i.e., pelvis and no-support. However, only 12/26 and 5/26 children in the SCI group were tested for these two levels, respectively. This activation in both, abdominal wall and thoracic paraspinals increases trunk stiffness to provide stability for an upright posture. Compared to thoracic paraspinal muscles, abdominal wall muscles are innervated by lower spinal segments, ranging from T7-L1 [29]. The higher activation of paraspinal and abdominal wall muscles above and below injury levels in children with SCI reflects the residual supraspinal influence on spinal motor neurons supplying these postural muscles. The identification of residual or preserved muscle activity below injury level in all participants with SCI regardless of the level, etiology or severity of injury provides a scientific basis for refining the current clinical practices and expectation for neuromuscular capacity and trunk control in children with SCI.

### 4.3. Activity Level in Upper Chest Muscles

Pectoralis and external intercostal muscles are considered to have dual function for posture control and respiration [30,31]. During various levels of support, children in both groups, produced activation in these two muscles. Both groups produced the same level of activation for all, except at below-ribs support level, where children in the SCI group produced higher activation in the pectoralis muscle than the TD group. This increased muscle activation may be due to forward arm movement during the test.

SCI-induced trunk muscle paralysis during rapid skeletal growth significantly hastens the onset and progression of scoliosis with nearly 100% of children injured before 10 developing scoliosis and 65% requiring surgery [10,11]. Given these significant functional impairments with added risk for developing scoliosis, understanding the effect of SCI on trunk muscles should be one of the primary focuses in rehabilitation research to change the trajectory of outcomes in children with SCI. Studies in adults with chronic SCI have shown evidence of muscular adaptations ranging from change in fiber composition, contractile properties to more fatigable. Pathological changes, such as muscle asymmetry and fiber composition in paraspinal muscles are thought to contribute to the progression of scoliosis in children with muscular dystrophies and adolescent idiopathic scoliosis. However, in contrast to cell-based therapeutic approaches [32], we propose that therapists may take advantage of the residual activation of trunk muscles below the lesion in children with SCI to improve posture control. We are in early stages of understanding the muscle contribution to development of scoliosis in children with SCI. Further research in children with SCI is warranted to explore and identify specific musculoskeletal adaptions and optimal therapeutic strategies to influence these adaptations. Therefore, assessment and quantification of trunk muscle activity using EMG is a critical means to understand the involvement and severity of muscle impairment after pediatric-onset SCI. In this study, we found that despite the obvious impairment in trunk control based on SATCo scores, children with SCI had activation of trunk muscles above and below the injury level during SATCo test. This observation has significant clinical implications as it can inform the development of evidence-based therapies for children with SCI capitalizing on a substantial amount of remaining neuromuscular capacity. The results of this study also support the need to revisit the current gold standard classification system assessing trunk muscle impairment based on the assessment of sensory function. In this study, we did not control for injury demographics such as etiology of injury, time since injury, severity of injury, intervention received (if any), and length of intervention received. These factors may have influenced the EMG activity observed during this study. Information on injury site and severity using neuroimaging techniques may be relevant to the presence of residual muscle activity below the lesion. In the current study, EMG activity was collected only during steady state control; however, anticipatory and reactive control testing may provide greater postural control challenge and thus additional knowledge on underlying postural muscle activity post SCI.

## 5. Conclusions

In children with SCI, activation of trunk muscles above and below injury, irrespective of injury level is indicative of residual or preserved supraspinal influence on spinal motor circuitry supplying these postural muscles. The evidence of this residual muscle activation below the injury level may provide opportunities to quantify and utilize this activation to promote upright sitting and balance required for functional activities in the pediatric population.

## Figures and Tables

**Figure 1 children-07-00278-f001:**
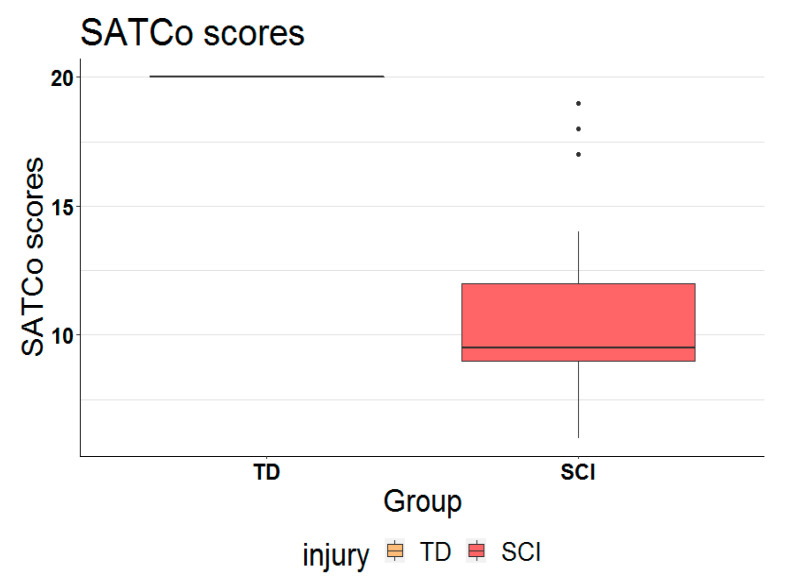
Comparison of Segmental assessment of trunk control (SATCo) scores between typically developing (TD) children and children with spinal cord injury (SCI). Note: every participant in the TD group scored 20/20 on SATCo test. The three dots (•••) represents outliers outside 95% cutoff.

**Figure 2 children-07-00278-f002:**
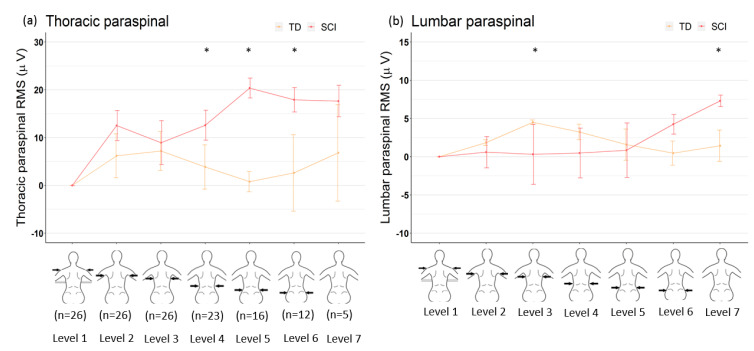
Comparison of trunk muscle activation during different SATCo support levels between children with spinal cord injury and typically developing children. Electromyography (EMG) data is represented as median of root mean square (RMS) values and standard error of median (SEM) bars of trunk muscles. (**a**) Thoracic paraspinal (PST), (**b**) Lumbar paraspinal (PSL), (**c**) Rectus abdominus (RA), (**d**) External oblique (OB), (**e**) Pectoralis major (PEC), and (**f**) External intercostal (INT) muscle.1 = shoulder level, 2 = Axilla, 3 = inferior scapula, 4 = over lower ribs, 5 = below ribs, 6 = pelvis, 7 = no support. * Indicates significant differences in root mean square values between the two groups.

**Figure 3 children-07-00278-f003:**
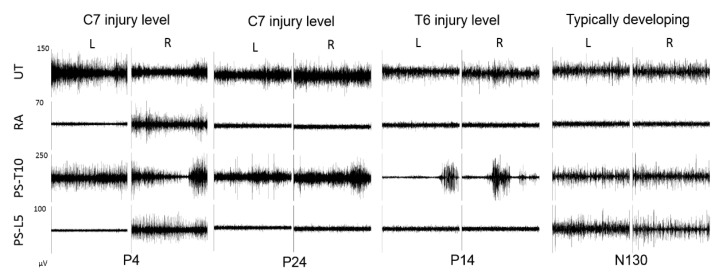
Trunk muscle activation in one typically developing (TD) child and three children with spinal cord injury (SCI) during SATCo test with over lower ribs support level. Note: activation of paraspinal muscles in children with SCI. UT = upper trapezius; RA = rectus abdominus; PS-T10 = thoracic paraspinal; PS-L5 = lumbar paraspinal muscle.

**Table 1 children-07-00278-t001:** Demographic characteristic of children with spinal cord injury and typically developing children.

ID	Age (y)	Sex	BMI	Injury/AIS Level	SATCo Score	Injury Etiology	Age at the Injury (y)	Time Since Injury (y)
**A85**	2	M	15.8	T10	19	Epidural abscess	1	2
**P24**	2	M	15.5	C6-T6	11	Epidural hematoma	1.6	1
**P33**	2	M	18.4	C3	8	Neuroblastoma	0	2
**P39**	2	F	18.2	T6	12	Neuroblastoma	0.8	1.8
**P12**	3	M	14.1	L1-2	12	Cardiac infarction	1.7	1.3
**P23**	3	M	13.9	C4-C7	8	Transverse Myelitis	2.4	1
**P34**	3	M	17.5	T4	7	Spinal astrocytoma	1.2	2.2
**P15**	4	F	15.0	T12	18	Epidural abscess	3	1
**P22**	4	M	15.5	T4	9	Motor vehicle accident	3.5	0.5
**P3**	4	M	16.1	T2	6	Neuroblastoma	0	4
**P8**	4	F	26.3	C2	9	Motor vehicle accident	1.3	3.4
**P14**	5	M	15.4	C7	9	Motor vehicle accident	3.5	2.3
**P16**	5	M	15.5	T11	18	Pedestrian hit by car	4	1
**P25**	5	F	15.4	T2	14	Spinal cord infarct	4.2	1
**P32**	5	F	18.0	T2-T3	14	Motor vehicle accident	3.3	1.4
**P13**	6	M	15.3	T3A	9	Motor vehicle accident	5.8	0.8
**P20**	6	M	25.1	T10	12	Motor vehicle accident	4.6	1.9
**P4**	6	F	13.6	C7-B	9	Surgical complication	6.4	0.3
**P40**	6	M	16.8	C2-D	9	Motor vehicle accident	4.7	1.7
**P7**	6	F	15.3	C4-C	9	Fall	3.5	2.5
**P21**	7	F	13.5	T10-A	17	Performing backbend	5.6	1.8
**P6**	7	M	15.5	T8-A	10	Motor vehicle accident	4.4	3.5
**P9**	7	F	17.7	T2-B	11	Motor vehicle accident	4	2.5
**P1**	9	M	14.6	T1-B	6	Epidural hematoma	4.2	4.6
**P10**	9	M	15.3	C5-C	12	Transverse Myelitis	0.4	8.9
**P30**	13	M	16.2	C5-A	4	Motor vehicle accident	2	11
**Mean ± SD**	5 ± 2	17M, 9F	16.5 ± 3	10C,15T,1L	10.8 ± 3	-	2.7 ± 1	2.7 ± 2
**P38**	3	F	15.0	NA	20	-	-	-
**N150**	4	M	15.9	NA	20	-	-	-
**N133**	4	M	18.0	NA	20	-	-	-
**N130**	5	F	16.4	NA	20	-	-	-
**N134**	5	M	15.1	NA	20	-	-	-
**N126**	6	M	16.7	NA	20	-	-	-
**N145**	6	F	9.2	NA	20	-	-	-
**N110**	7	F	18.0	NA	20	-	-	-
**N147**	9	F	18.0	NA	20	-	-	-
**N109**	12	M	17.2	NA	20	-	-	-
**Mean ± SD**	6 ± 2	5M, 5F	15 ± 2	NA	20	-	-	-

BMI = Body mass index, SATCo = Segmental assessment of trunk control, T = thoracic, C = cervical, L = lumbar. AIS = American spinal injury association impairment scale (If age <7 years, the injury level listed in medical records was reported).

**Table 2 children-07-00278-t002:** Summary of categorical variables of typically developing children and children with spinal cord injury.

	SCI	TD	*P*
	*n* = 26	*N* = 10	
Median (IQR)	5 (3,6)	5 (4,7)	0.35
Median (IQR)	15 (15,17)	16 (15,18)	0.54
Female, *n* (%)	9 (35%)	5 (50%)	-
Male, *n* (%)	17 (65%)	5 (50%)	-
Cervical, *n* (%)	10 (38%)	-	-
Thoracic, *n* (%)	15 (58%)	-	-
Lumbar, *n* (%)	1 (4%)	-	-

SCI = Spinal cord injury, TD = Typically developing, IQR = Interquartile range (min, max).

**Table 3 children-07-00278-t003:** Root mean square (RMS) values of trunk muscles during SATCo assessment in typically developing children and in children with spinal cord injury.

		SCI	TD	
		26	10	
Muscle	Measurement			*p*-value
PEC	Axilla (median (IQR))	−5 (−11,3)	−1 (−6,0)	0.82
	Inferior scapula (median (IQR))	−1 (−9,3)	1 (0,5)	0.53
	Over lower ribs (median (IQR))	−2 (−8,14)	1 (−1,2)	0.80
	Below ribs (median (IQR))	11 (6,30)	0 (−2,3)	0.002 *
	Pelvis (median (IQR))	15 (−1,66)	0 (−4,2)	0.23
	No support (median (IQR))	35 (0,70)	0 (−4,1)	0.11
INT	Axilla (median (IQR))	22 (10,32)	26 (10,77)	0.42
	Inferior scapula (median (IQR))	15 (7,29)	30 (23,57)	0.04 *
	Over lower ribs (median (IQR))	13 (0,26)	21 (13,63)	0.08
	Below ribs (median (IQR))	3 (−6,36)	20 (5,73)	0.12
	Pelvis (median (IQR))	15 (−1,57)	38 (13,111)	0.13
	No support (median (IQR))	11 (3,26)	24 (10,37)	0.39
RA	Axilla (median (IQR))	1 (0,2)	0 (0,1)	0.10
	Inferior scapula (median (IQR))	1 (0,1)	0 (0,1)	0.48
	Over lower ribs (median (IQR))	0 (0,3)	0 (0,1)	0.19
	Below ribs (median (IQR))	3 (0,4)	0 (0,0)	0.02 *
	Pelvis (median (IQR))	2 (1,3)	0 (0,0)	0.003 *
	No support (median (IQR))	1 (1,8)	0 (0,0)	0.007 *
OB	Axilla (median (IQR))	0 (0,1)	0 (−3,0)	0.08
	Inferior scapula (median (IQR))	0 (0,1)	0 (−3,0)	0.38
	Over lower ribs (median (IQR))	0 (−2,1)	0 (−1,1)	0.69
	Below ribs (median (IQR))	2 (−2,5)	0 (0,1)	0.36
	Pelvis (median (IQR))	5 (1,12)	0 (−1,1)	0.002 *
	No support (median (IQR))	7 (7,21)	0 (−1,1)	0.009 *
PST	Axilla (median (IQR))	13 (0,21)	6 (1,10)	0.41
	Inferior scapula (median (IQR))	9 (1,22)	7 (0,17)	0.72
	Over lower ribs (median (IQR))	13 (5,29)	4 (−2,7)	0.01 *
	Below ribs (median (IQR))	20 (15,24)	1 (−1,4)	0.001 *
	Pelvis (median (IQR))	18 (4,33)	3 (0,6)	0.03 *
	No support (median (IQR))	18 (15,35)	7 (2,14)	0.06
PSL	Axilla (median (IQR))	1 (0,2)	1 (−1,5)	0.77
	Inferior scapula (median (IQR))	0 (0,4)	5 (1,15)	0.03 *
	Over lower ribs (median (IQR))	0 (0,7)	3 (0,11)	0.32
	Below ribs (median (IQR))	1 (0,10)	2 (0,11)	0.67
	Pelvis (median (IQR))	4 (1,8)	0 (−1,2)	0.07
	No support (median (IQR))	7 (7,9)	1 (0,3)	0.004 *

SCI = Spinal cord injury, TD = Typically developing, PEC = Pectoralis major, INT = External intercostal, RA = Rectus abdominus, OB = External oblique, PST = Thoracic paraspinal, PSL = Lumbar paraspinal. IQR = Interquartile range (min, max), * Indicates significant differences in root mean square values between the two groups.

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
