# Peer review of "Contribution of Trunk Muscles to Upright Sitting with Segmental Support in Children with Spinal Cord Injury"

_children, 2020, doi:10.3390/children7120278_

Round 1

Reviewer 1 Report

The authors performed a very interesting study on the impact of muscle function on the outcome of children with an injury of the upper spinal cord. I have only a few suggestions for the authors to consider:

  • The authors self-cited the SATCo scale (reference nr 20), but it is not a very relevant paper, and I would rather replace it with another better presenting the value of this scale (PMID: 28749721).
  • They provided a very interesting observation on the ubiquitous development of scoliosis. The authors very precisely described pathology, but there is a missing therapeutic perspective. Therefore, I suggest discussing the authors’ findings against the regenerative approach to the muscular contribution to scoliosis (PMID: 31170887).
  • I would also discuss the muscular changes following spinal cord injury to provide a better molecular landscape to other therapeutic opportunities (PMID: 11838582).

Reviewer 2 Report

Discuss explanation for trunk muscles working below the level of injury.

It appears that's more hypertonia and spinal reflexes than preserved upper motor pathways.
